# Biosafety devices to control the spread of potentially contaminated dispersion particles. New associated strategies for health environments

**Victor Angelo Martins Montalli**[1]*, **Patrícia Rejane de Freitas**[1], **Milenna de Figueiredo Torres**[1], **Oscar de Figueiredo Torres Junior**[2], **Dienne Hellen Moutinho De Vilhena**[1], **José Luiz Cintra Junqueira**[1], **Marcelo Henrique Napimoga**[1]

**1** Faculdade São Leopoldo Mandic, Instituto São Leopoldo Mandic, Campinas–SP, Brazil, **2** UVCtec Company, São Paulo—SP, Brazil

☯ These authors contributed equally to this work.
* victor.montalli@slmandic.edu.br, victormontalli@gmail.com

**Data Availability Statement:** All relevant data are within the manuscript and its Supporting Information files.

**Funding:** CAPES #001.

## Abstract

Dental procedures produce a large amount of spatter and aerosols that create concern for the transmission of airborne diseases, such as Covid-19. This study established a methodology with the objective of evaluating new associated strategies to reduce the risk of cross-transmission in a health environment by simulating spread of potentially contaminated dispersion particles (PCDP) in the environment. This crossover study, was conducted in a school clinic environment (4 clinics containing 12 dental chairs each). As a positive control group (without barriers), 12 professionals activated at the same time the turbine of dental drill, for one minute, with a bacterial solution *(Lactobacillus casei* Shirota, $1.5 \times 10^8$ CFU/mL), which had been added in the cooling reservoir of the dental equipment. In the experimental groups, the professionals made use of; a) an individual biosafety barrier in dentistry (IBBD) which consists of a metal support covered by a disposable PVC film barrier; b) a Mobile Unit of Disinfection by Ultraviolet-C, consisting of 8 UV lamps-C of 95W, of 304μW/cm² of irradiance each, connected for 15 minutes (UV-C) and; c) the association between the two methods (IBBD + UV-C). In each clinic, 56 Petri dishes containing MRS agar were positioned on the lamps, benches and on the floor. In addition, plates were placed prior to each test (negative control group) and plates were also placed in the corridor that connects the four clinics. In the groups without barrier and IBBD + UV-C the passive air microorganisms in Petri dishes was also evaluated at times of 30, 60, 90 and 120 minutes after the end of the dental's drill activation. The mean (standard deviation) of CFU of *L. casei* Shirota for the positive control group was 3905 (1521), while in the experimental groups the mean using the IBBD was 940 (466) CFU, establishing a reduction on average, of 75% (p<0.0001). For the UV-C group, the mean was 260 (309) CFU and the association of the use of IBBD + UV-C promoted an overall average count of 152 (257) CFU, establishing a reduction on average of 93% and 96%, respectively (p<0.0001). Considering these results and the study model used, the individual biosafety barrier associated with UV-C technology showed to be efficient strategies to reduce the dispersion of bioaerosols generated in an environment with

**Competing interests:** The authors have not declared any conflict of interest. UVCtec Company is a startup commercial entity in the UV-C field, but has no interest in any of the equipment used in the present study, only in contributing its network of contacts towards the present study, in order to better understand UV-C, so that dental clinics, hospitals, health practices, labs and associated dental and medical supply chain smaller businesses can remain open and operate safely through any future viral pandemics. Mr. Oscar Torres-Junior has support in the form of salary from UVCtec, although this does not alter our adherence to PLOS ONE policies on sharing data and materials.

high rate of PCDP generation and may be an alternative for the improvement of biosafety in different healthy environment.

## Introduction

Most dental treatments are aerosol-generating procedures (AGPs) that produce a mixture of spatter, drops and aerosols containing saliva, blood, irrigating water, and viable microorganisms (including bacteria, fungi, and viruses) [1]. Commonly used dental instruments, including dental handpieces and ultrasonic equipment, generate a large potentially contaminated dispersion particles (PCDP), which pose a risk to professionals and patients [2, 3]. These microparticles are invisible, therefore mapping their spatial distribution within the clinical environment is neglected, consequently developing better ways to mitigate the risk of disease transmission is of great importance.

The PCDP generated during the appointments can remain in the air for less time (droplets, 5–100μm) or longer (aerosols, ≤5μm) and these fall on the surfaces of the environment under the influence of gravity, following a ballistic trajectory from the point of origin. In addition, droplets can remain suspended in the air until the water evaporates, and aerosols can remain suspended for several hours and can flatten for meters from their source of origin [1, 4, 5].

Much more attention was focused on dental aerosol generating procedures (AGPs) because of Covid-19 [6]. In some cases, especially when people are close to each other, it has been proven that Covid-19 spreads by aerial transmission [7].

Among microorganisms that are potentially contagious to health professionals operating near the face and oral cavity, especially when PCDP is generated [8], are hepatitis B virus, HIV (human immunodeficiency virus) as well as SARS-CoV-2 (Severe Acute Respiratory Syndrome coronavirus 2). The latter can remain infectious in aerosols for long periods, even when water evaporates, and particles that settle on surfaces can remain infectious for up to 72 h [9, 10]. It is of considerable interest to have methods to reduce the dispersion of splashes/droplets/aerosols during procedures. In a preliminary study, the Individual Biosafety Barrier in Dentistry (IBBD), which is a biosafety device, was tested aiming to reduce the dispersion of droplets and aerosols generated during the service, reducing the CFU count by 95% [11]. Other studies have also used bacteria colony counts [3, 12] and other fluorescent tracers to show the distribution of the ejected material in general [13, 14].

While health environments are cleaned and disinfected regularly by the use of manual techniques, evidence suggests that the adequacy of cleanliness is often suboptimal, particularly when the focus is only on surfaces perceived as high risk or frequently touched [15]. Inadequate cleaning using manual techniques led to the development of no-touch systems that can decontaminate objects and surfaces in the patient's environment [16], among these technologies are those that employ ultraviolet (UV) light [17, 18].

Automated UV disinfection devices that continuously emit UV-C in the 254 nm wavelength range have been used in health environments with the aim of decontaminating the environment. Some of these systems can reduce by up to 4 log the microbial load of the environment [19].

However, there are no established efficacy standards for UV devices which has resulted in manufacturers using different approaches for such UV disinfection devices. This lack of standardization created confusion in the health sector. Currently, infection prevention experts cannot accurately compare the performance of UV devices and make purchasing decisions. Also,

without a pattern of effectiveness, users are unable to follow any revalidation protocol for continuous device effectiveness [20].

Ultraviolet disinfection technology can be used to supplement manual cleaning, and recently it has become an acceptable method of no-touch disinfection within healthcare facilities and is currently routinely used in disinfecting the hospital environment with hospital-acquired infection reduction having been demonstrated in previous studies [15–20].

Therefore, the aim of this study was to map at defined distances the distribution of PCDP in a university dental care clinic. In addition, using the microbial dispersion model, we propose methods for dispersion control, making use of the individual biosafety barrier (mechanical method) and UV-C technology (physical method) as well as the association of both methods for contamination control during high microbial dispersion model.

## Material and methods

This research was approved by the Ethics and Research Committee of Faculdade São Leopoldo Mandic, Campinas, SP, Brazil (2020–0603) and was conducted at Faculdade São Leopoldo Mandic (Campinas, SP, Brazil). The clinical part was carried out in the post-graduate clinic building, with 144 dental equipment, distributed over three floors. Each floor consists of 4 clinics There are 12 dental equipment (Dabi Atlante®, Ribeirão Preto, SP, Brazil) in each clinic (12 m x 6.85 m x 2.5 m) positioned at a linear distance of 2.0 m from each other, 6 on the right side and 6 on the left side (S1 Fig). Four ground-floor clinics were used. The dental clinic used for this study was closed to the public during the experiment, i.e. no patients were present, all doors and windows were kept shut to prevent air draft and the air conditioning system was off throughout the experiment.

Twelve undergraduate and graduate students of the Dental course were invited to participate in the research. Each participant was previously instructed about the tests and each received an identification and was positioned in the same dental chair position in the 4 different clinics. In addition, all activated dental drill at the same time and positioned on the right side of work. To simulate a clinical situation of cavity preparation, a diamond tip was added to the dental drill which was activated on a stock tooth for one minute. After activation, the Petri dishes were opened and remained open for 15 minutes in the pre stablished position. S2 Fig demonstrates each of the tested environments.

### Individual biosafety barrier in dentistry

This protection barrier against droplets and aerosol is made using a metal support, with a 30 cm ring and covered by a disposable 30 microns thickness PVC film measuring approximately 1.5 x 1.5 m (patent required BR 20 2020 019471 8) installed in the activation region of dental drill (S3 Fig).

### Ultraviolet-C device mobile disinfection unit

The tested equipment (UMDUV 2.0) is produced by the UVCtec Company (São Paulo, SP, Brazil) composed of 8 UVGI lamps of 95W of low-pressure mercury with 304µW/cm$^2$ of irradiance, without ozone generation (S3 Fig). It is remotely controlled by a smartphone application and a multiprocessed circuit capable of programming the time needed to deactivate the most different organisms found today, which can be bacteria, viruses, spores and fungi, in various environments. Controlled remotely by bluetooth, direct contact of the operator with the equipment is eliminated, there are also presence of sensors that cover 360˚ in its surroundings, turning it immediately off in the presence of any movement within a radius to up to 9 meters from the issuing source, which gives it operational safety. The equipment was positioned in the center of the clinics in the UV-C and IBBD + UV-C groups and after opening the Petri

dishes and with no person in the room, the device was switched on for 15 minutes. Then, the participants returned, each in their dental chair, to the closing of the dishes.

## Generation of Potentially Contaminated Dispersion Particles (PCDP)

In summary, bacterial suspensions containing the microorganism *Lactobacillus casei* Shirota (Yakult Brasil Ltda, Lot # 0818F1139) were used in the experiments. This strain was chosen because it is a bacterial species that poses no risk of environmental contamination and measures 0.5 μm (the SARS-CoV-2 virus measures 0.1 μm). Additionally, this microorganism has already been tested and validated for the dispersion model in a dental clinic environment in previous studies [2, 11]. Thus, a viability test was performed to determine the initial concentration of $1.5 \times 10^8$ CFU / mL of *L. casei* Shirota.

Microbiologic microbial growth tests were performed using lactobacillus spp. enriched agar (DeMan, Rogosa and Sharpe, MRS, Merck). After the collections, the samples were incubated in an incubator at 37°C for 48 hours in aerobiosis.

The water container to be used in the cooling of the high-rotation turbine, received *the Lactobacillus casei* Shirota solution at a concentration of $1.50 \times 10^8$ CFU/mL.

Petri dishes were placed with MRS on the right and left sides of the clinic, on the surface: of the benches (n = 12), of the dental cart (n = 12), of the auxiliary tables (n = 6); at the top, by the ceiling fluorecent lamps fixture (n = 14) and; on the floor, below the headsupport of each dental chair (n = 12), totaling 56 plates per clinic (S1 Fig). The positive control group consisted of the activation of the high-speed turbine for 1 minute by the 12 participants at the same time without any barrier. Then, the Petri dishes were opened by each volunteer shortly after the activation of the dental drill and remained open for 15 minutes.

Colony Forming Units (CFU) were counted and Gram staining was performed to confirm the Lactobacilli culture. The tests were performed in triplicate. The size of the Petri dishes was 90 mm in diameter and the area 63.62 cm$^2$. Petri dishes which contained less than 300 CFU of *Lactobacillus casei* Shirota were counted in full. Petri dishes containing myriads of CFU had colonies counted based on three areas measuring 1 cm$^2$ each [2, 21]. Then, the average was calculated and multiplied by 63.62 (total area of the Petri dish). CFUs were counted manually (aided by a CFU counter).

## Evaluation of UV-C device dosimetry

The radiometer used in this work was the MRUR-203, a 254nm short wave ultraviolet radiation meter (UV-C), with UV sensor with correction filter with selected sensitivity range of 1.999mW/cm$^2$. The radiometer was operated according to its specification at room temperature of 22°C at 53% of air relative humidity. The UV sensor was placed where the Petri dishes were positioned (floor, surfaces, and fluorecent lamp fixture, at different distances).

The lamp entry time was first evaluated by measuring irradiance (units of milliWatt per square centimeter, mW/cm$^2$) as a function of time repeatedly from cold start. Then, irradiance was measured in each of the sensor positions, recording measurements at intervals of 30 s up to 180 s, and taking the mean, which was considered as irradiance received by point. The amount of fluency received at each point was considered by the total application time, which was 15 minutes.

## Dose of UV-C exposure (fluency)

The microorganisms exposed to UV-C irradiation are subject to a dose of exposure (fluency) which is a function of irradiance multiplied by the exposure time, noting that the irradiance decreases with the inverse of the square of the distance, as follows:

$\varphi = t \cdot EuR$

$\varphi$ = dose of exposure to UV-C (fluency), $J/m^2$

t = exposure time, sec

EuR = Irradiance, $W/m^2$

### Thermal performance of UV-C device

The thermal performance of the equipment was investigated using images from a FLIR camera (Model: FLIR-E49001). This camera was used to capture thermal images from the UMDUV device and record temperature for 60 seconds.

### Density of PCDP dispersed in the dental clinics

With the aid of an analytical scale (model M214Ai, BEL, Monza, Italy) a wrapper was weighed (Segplast, São Paulo, Brazil), measuring 5 cm x 23 cm (individual average weight of 0.92g). To evaluate the total weight (in grams) that the high-speed turbine releases in 1 minute, 3 weighing were performed at 4 different times. The average weight dispensed by dental drill in one minute was 71.01g. Subtracting the weight from the package (0.92g), the weight of 70.09g of the total weight was used as the basis.

Each volunteer collected the liquid that was dispensed in a plastic bottle after the high-speed turbine was driven for 1 minute. Each wrapper was identified with the number of the respective dental equipment and in which experimental group corresponded the package. The wrappers were stored and weighed later to determine whether the volume used by each volunteer was close.

### Evaluation of PCDP suspension in the dental clinical environment

In the dental clinics of the barrier-free group and the IBBD + UV-C group, a support table was positioned in the center of the clinic. Petri dishes with MRS agar were placed, which were opened at 30, 60, 90 and 120 minutes after the end of the activation of the high-rotation turbine, in triplicate.

### Evaluation of the presence of PCDP outside the clinical setting

A corridor joins the entrance doors of the clinics. In the clinic without the use of barriers, after the end of the activation of dental drill by the volunteers, Petri dishes containing MRS agar were positioned. Using as reference the entrance door, they were directed at distances of 1, 5 and 10 meters and remained open for 20 minutes.

### Statistical analysis

Data from both experiments were examined for normality by the Shapiro-Wilk test. As data demonstrated normality, all analyses were then performed using parametric methods. The differences in CFU for the different distances were compared by One-Way ANOVA, followed by Tukey's test. To evaluate the dispersion time of CFU, the Two-Way ANOVA, followed by Bonferroni's multiple comparisions. The level of significance was established at 5%. All statistical analyses were performed with GraphPad Prism v8.0.

## Results

The predominant environmental conditions were 22°C and 53% relative humidity throughout the experiments.

From the Petri dishes arranged in dental clinics before the beginning of the tests, the control group had low *Lactobacillus casei* Shirota CFU counts. The group without barriers had mean (standard deviation) of 1.3 (1.0) CFU; the IBBD group 11.3 (6.1); the UV-C group 1.0 (0.8) and in the IBBD + UV-C group 2.3 (2.1), with no statistically significant difference (p>0.05) between the dental clinics used for the experiments.

To quantify the mass (in grams) of the bacterial solution dispersed in the environment, the solution dispensed at the time of dental drill refrigeration was collected from each volunteer and subtracted by 70.9 (g), allowing to quantify, passively, the amount of PCDP that were generated in each environment. From each of the 12 volunteers, 35.1 (19.2) g was dispersed in the environment without barriers. In the IBBD clinic, the mean (SD) of dispersed mass was 34.1 (16.8) g per volunteer. In the UV-C clinic, the mean (SD) was 43.1 (9.6) g and in the IBBD + UV-C clinic, the value was 38.7 (19.1) g mean (SD). There was no statistically significant difference between the mass of PCDP (in grams) dispersed in the clinics (p >0.05) (S4 Fig).

In Fig 1A–1D it is possible to observe the box plot of the data of the CFU counts as well as a heat map of the dispersion of the counts of the plates on the different surfaces analyzed. Fig 1A shows the data of all surfaces analyzed in a grouped manner; fluorescent lamps fixture (Fig 1B); in the stands (Fig 1C) and on the floor (Fig 1D) and all CFU counting data are summarized in Table 1 and described briefly below. Regarding the CFU count and PCDP dispersion (considering the Petri dishes opened for 15 minutes after the activation of dental drill in the 12 dental chairs), in the control group (without barrier), the minimum and maximum value of CFU counted was 1527 and 7613 with and mean (standard deviation) of 3905 (1521) CFU

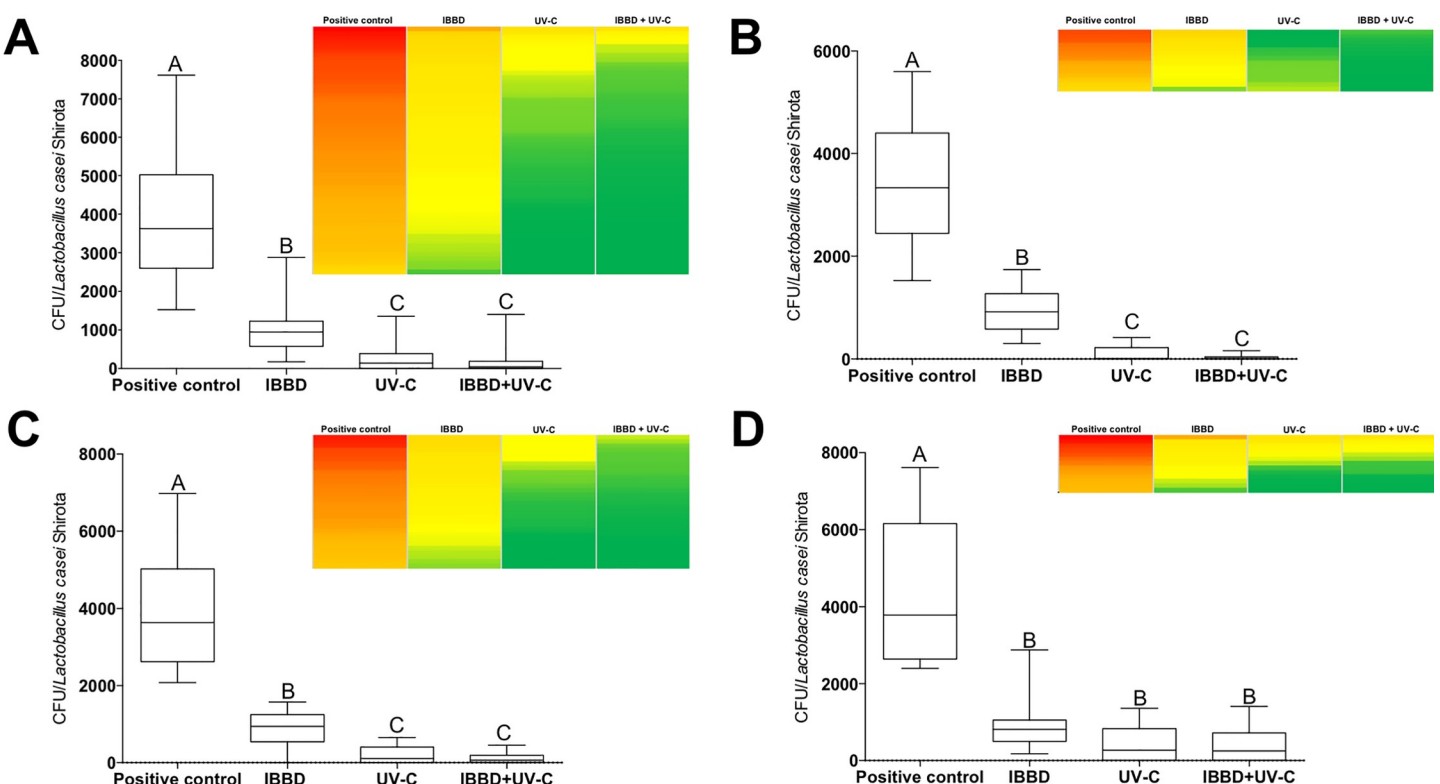

**Fig 1. Box plot of the data of the CFU counts and a heat map of the dispersion of the counts of the plates on the different surfaces analyzed.** Data of all surfaces analyzed in a grouped manner (A); fluorescent lamp fixtures (B); in the stands (C) and on the floor (D). Different letters indicate statistical significance (p < 0.0001). One-way ANOVA and post hoc analysis using Tukey's multiple comparation.

**Table 1. CFU counts and a heat map of the dispersion of the counts of the plates on the different surfaces and groups analyzed.**

| | | Positive control | | IBDB | | UV-C | | IBDB + UV-C | | Reduction comparing with positive control (%) | | |
|---|---|---|---|---|---|---|---|---|---|---|---|---|
| Chair number | Plate area | CFU | Heat map | CFU | Heat map | CFU | Heat map | CFU | Heat map | IBDB | UV-C | IBDB + UV-C |
| 12 | STAND 13 | 5450 | | 1166 | | 652 | | 128 | | 79 | 88 | 98 |
| 12 | STAND 14 | 6680 | | 1569 | | 624 | | 202 | | 77 | 91 | 97 |
| 12 | LF 7 | 5599 | | 1591 | | 361 | | 47 | | 72 | 94 | 99 |
| 12 | FLOOR 7 | 7104 | | 1072 | | 1357 | | 352 | | 85 | 81 | 95 |
| 11 | STAND 11 | 5641 | | 1548 | | 624 | | 452 | | 73 | 89 | 92 |
| 11 | STAND 12 | 6532 | | 1251 | | 0 | | 198 | | 81 | 100 | 97 |
| 11 | STAND 30 | 6977 | | 1244 | | 285 | | 266 | | 82 | 96 | 96 |
| 11 | LF 6 | 5280 | | 1230 | | 174 | | 160 | | 77 | 97 | 97 |
| 11 | LF 15 | 5440 | | 1410 | | 268 | | 104 | | 74 | 95 | 98 |
| 11 | FLOOR 6 | 5874 | | 948 | | 806 | | 712 | | 84 | 86 | 88 |
| 10 | STAND 15 | 4347 | | 1272 | | 267 | | 137 | | 71 | 94 | 97 |
| 10 | STAND 16 | 3732 | | 1166 | | 123 | | 39 | | 69 | 97 | 99 |
| 10 | LF 8 | 4241 | | 1739 | | 272 | | 0 | | 59 | 94 | 100 |
| 10 | FLOOR 8 | 7613 | | 2884 | | 1271 | | 142 | | 62 | 83 | 98 |
| 9 | STAND 9 | 4220 | | 1230 | | 219 | | 1 | | 71 | 95 | 100 |
| 9 | STAND 10 | 5577 | | 1527 | | 263 | | 65 | | 73 | 95 | 99 |
| 9 | STAND 29 | 5026 | | 852 | | 63 | | 44 | | 83 | 99 | 99 |
| 9 | LF 5 | 3987 | | 920 | | 276 | | 61 | | 77 | 93 | 98 |
| 9 | FLOOR 5 | 6256 | | 1000 | | 10 | | 136 | | 84 | 100 | 98 |
| 8 | STAND 17 | 2905 | | 997 | | 61 | | 25 | | 66 | 98 | 99 |
| 8 | STAND 18 | 3817 | | 1421 | | 2 | | 0 | | 63 | 100 | 100 |
| 8 | LF 9 | 2693 | | 1145 | | 1 | | 0 | | 57 | 100 | 100 |
| 8 | FLOOR 9 | 4708 | | 1108 | | 149 | | 10 | | 76 | 97 | 100 |
| 7 | STAND 7 | 3414 | | 912 | | 5 | | 186 | | 73 | 100 | 95 |
| 7 | STAND 8 | 5026 | | 992 | | 4 | | 17 | | 80 | 100 | 100 |
| 7 | STAND 28 | 3923 | | 968 | | 0 | | 0 | | 75 | 100 | 100 |
| 7 | LF 4 | 4453 | | 956 | | 8 | | 2 | | 79 | 100 | 100 |
| 7 | FLOOR 4 | 4178 | | 956 | | 5 | | 7 | | 77 | 100 | 100 |
| 6 | STAND 19 | 2651 | | 1103 | | 76 | | 0 | | 58 | 97 | 100 |
| 6 | STAND 20 | 2545 | | 1272 | | 2 | | 0 | | 50 | 100 | 100 |
| 6 | LF 10 | 2651 | | 572 | | 9 | | 2 | | 78 | 100 | 100 |
| 6 | FLOOR 10 | 3202 | | 676 | | 1 | | 1 | | 79 | 100 | 100 |
| 5 | STAND 5 | 4050 | | 800 | | 39 | | 98 | | 80 | 99 | 98 |
| 5 | STAND 6 | 4284 | | 1004 | | 2 | | 0 | | 77 | 100 | 100 |
| 5 | STAND 27 | 3478 | | 904 | | 0 | | 0 | | 74 | 100 | 100 |
| 5 | LF 3 | 3711 | | 668 | | 6 | | 7 | | 82 | 100 | 100 |
| 5 | FLOOR 3 | 3393 | | 608 | | 66 | | 452 | | 82 | 98 | 87 |
| 4 | STAND 21 | 2333 | | 604 | | 178 | | 14 | | 74 | 92 | 99 |
| 4 | STAND 22 | 3181 | | 544 | | 87 | | 78 | | 83 | 97 | 98 |
| 4 | LF 11 | 1739 | | 1315 | | 274 | | 5 | | 24 | 84 | 100 |
| 4 | FLOOR 11 | 2587 | | 456 | | 27 | | 19 | | 82 | 99 | 99 |
| 3 | STAND 3 | 2418 | | 456 | | 139 | | 204 | | 81 | 94 | 92 |
| 3 | STAND 4 | 3542 | | 560 | | 83 | | 64 | | 84 | 98 | 98 |
| 3 | STAND 26 | 3520 | | 672 | | 94 | | 31 | | 81 | 97 | 99 |
| 3 | LF 2 | 2948 | | 596 | | 127 | | 24 | | 80 | 96 | 99 |

*(Continued)*

**Table 1.** (Continued)

| Chair number | Plate area | Positive control | | IBDB | | UV-C | | IBDB + UV-C | | Reduction comparing with positive control (%) | | |
|---|---|---|---|---|---|---|---|---|---|---|---|---|
| | | CFU | Heat map | CFU | Heat map | CFU | Heat map | CFU | Heat map | IBDB | UV-C | IBDB + UV-C |
| 3 | FLOOR 2 | 2820 | | 640 | | 517 | | 1406 | | 77 | 82 | 50 |
| 2 | STAND 23 | 2142 | | 412 | | 624 | | 22 | | 81 | 71 | 99 |
| 2 | STAND 24 | 2333 | | 348 | | 596 | | 376 | | 85 | 74 | 84 |
| 2 | LF 12 | 2375 | | 560 | | 420 | | 7 | | 76 | 82 | 100 |
| 2 | FLOOR 12 | 2566 | | 376 | | 388 | | 876 | | 85 | 85 | 66 |
| 1 | STAND 1 | 2078 | | 311 | | 388 | | 230 | | 85 | 81 | 89 |
| 1 | STAND 2 | 3181 | | 536 | | 444 | | 149 | | 83 | 86 | 95 |
| 1 | STAND 25 | 2375 | | 408 | | 612 | | 192 | | 83 | 74 | 92 |
| 1 | LF 1 | 1527 | | 304 | | 107 | | 33 | | 80 | 93 | 98 |
| 1 | LF 13 | 1951 | | 668 | | 264 | | 11 | | 66 | 86 | 99 |
| 1 | FLOOR 1 | 2396 | | 176 | | 840 | | 720 | | 93 | 65 | 70 |
| | Min | 1527 | | 176 | | 0 | | 0 | | 24 | 65 | 50 |
| | Max | 7613 | | 2884 | | 1357 | | 1406 | | 93 | 100 | 100 |
| | Mean | 3905 | | 940 | | 260 | | 152 | | 75 | 93 | 96 |
| | SD | 1521 | | 466 | | 309 | | 257 | | 11 | 9 | 9 |
| | p | A | B | C | C | - | - | - | | | | |

Absolute and relative values of CFU reduction compared to the positive control group. Maximum values, minimums, mean, standard deviation and inferential analysis is also presented. Different letters indicate statistical significance (p < 0.0001). One-way ANOVA and post hoc analysis using Tukey's multiple comparation.

(Fig 2A). For the group (IBBD), the minimum value was 176 and maximum 2884, with an average of 940 (466) CFU (Fig 2B), while in the UV-C group the minimum value was 0 and maximum of 1357, with an average of 260 (309) CFU (Fig 2C), and minimum 0 and maximum 1406 in the IBBD+UV-C group, with an average of 152 (257) CFU (Fig 2D). The mean difference between the control group and the IBBD group was, on average, 75%. When using UV-C technology, average CFU counts had, on average, a 93% and 96% reduction in IBBD +UV-C. In the analysis of variance, a significant difference was observed between the group without barriers and the other experimental groups (F (3,220) = 258.3, p = <0.0001). In addition, post hoc analysis using Tukey's multiple comparation criterion for significance indicated statistical difference between the IBBD and UV-C groups (p < 0.0001) and there was no statistically significant difference between the UV-C and IBBD+UV-C groups (p>0.05). To evaluate the distance that PCDP can reach from the generating source, new Petri dishes were positioned in the corridor that connect the clinics to the end of the group test without barriers. As shown in Fig 2E, it was observed the growth of 903 (28) CFU in the plates positioned 1m from the input (yellow box), 3044 (64) CFU at 5m distance (dark orange box) and 1966 (42) CFU 10 m away from the generating source (light orange box). Schematic 3D data are presented in S5 Fig.

Fig 3 shows the sedimentation time in the environments of the control group and the IBBD + UV-C group. After 30 minutes of activation of the dental drill containing *L. casei*, in the group without barriers the mean CFU was 3167 (435) CFU, while for the IBBD + UV-C group it was 5 (2) CFU (p<0.0001). After 60 minutes of activation, the mean suspended CFU that was deposited on the MRS board was 441 (13) in the group without barrier and 1(0) in the IBBD + UV-C group (p<0.0001). At 90 minutes after activation, the mean CFU of the group

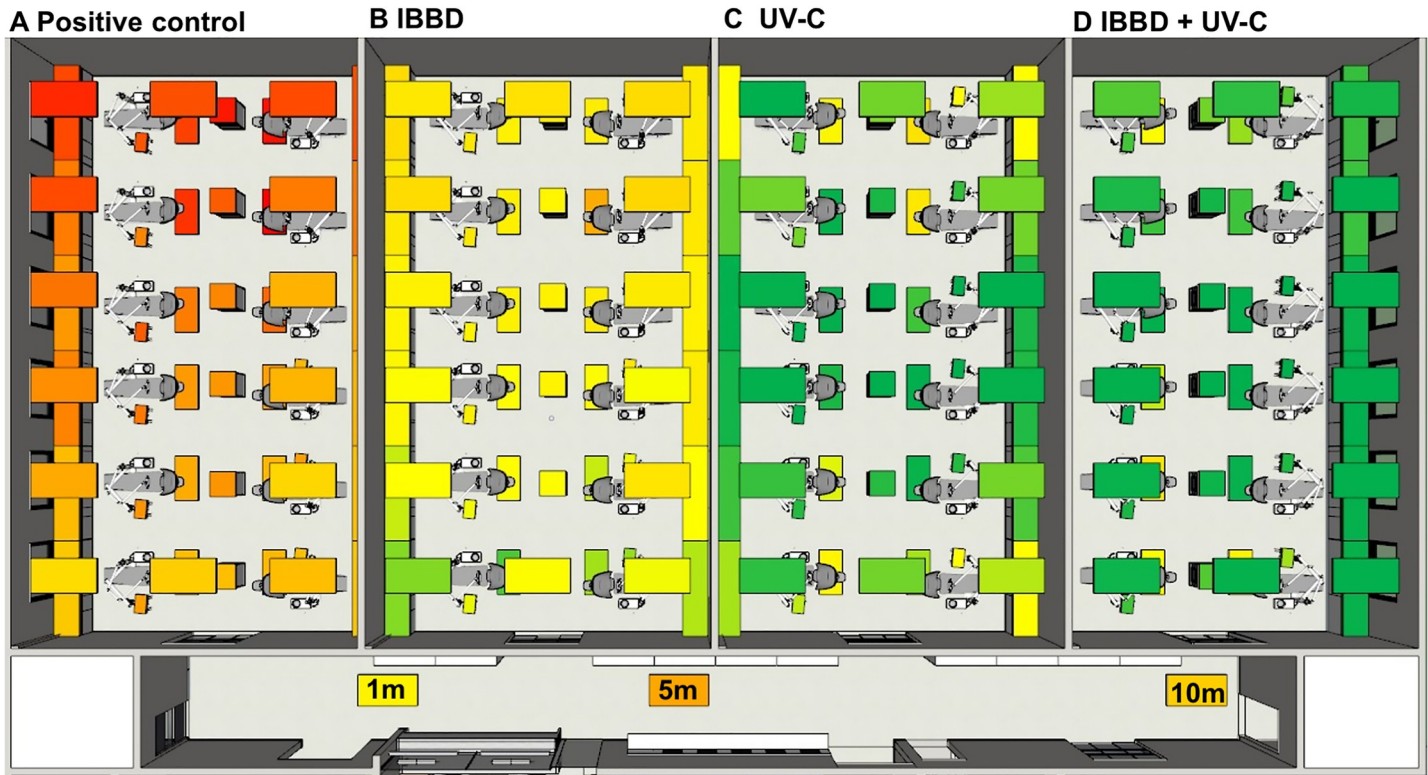

**A Positive control**    **B IBBD**    **C  UV-C**    **D IBBD + UV-C**

**1m**    **5m**    **10m**

**E Corridor**

**Fig 2.** Heat map of the results obtained from CFU counts in clinics: Positive Control (A); IBBD; UV-C; IBBD + UV-C and Corridor. The mean difference between the positive control group and the IBBD group was, on average, 75% (B). When using UV-C device, average CFU counts had, on average, a 93% (C) and 96% reduction in IBBD+UV-C (D). To evaluate the distance that PCDP can reach from the generating source, new Petri dishes were positioned in the corridor that connect the clinics to the end of the group test without barriers. It was observed the growth of CFU in the plates positioned 1m from the input, at 5m distance and 10 m away from the generating source (E).

without barrier was 121 (1) and 0 (0) in the IBBD + UV-C group (p>0.05). After 120 minutes of dental drill activation, the growth of 40 (8) CFU in the control group and 0 (0) in the IBBD + UV-C group (p>0.05) was observed.

A negative correlation between CFU and UV-C fluency ($J/m^2$) was achieved after 15 minutes of the equipment in operation, and a negative correlation (r = -0.62) can be observed in Fig 4A, indicating that the greater the amount of energy in the area, the more effective the bactericidal activity of the UV-C technology tested. Irradiance ($W/m^2$) and Fluency ($J/m^2$) data are presented in Fig 5. In addition, the temperature generated by the lamps with the UV-C equipment connected through the use of FLIR was evaluated, being observed the measured temperature of 98.8˚C, but without remarkable oscillation of heat generation around the equipment (Fig 4B) in the 4 clinical environments evaluated.

## Discussion

It is known that the health environments that generate droplets and aerosols have received special attention in sanitary measures, due to the risk of cross-contamination, especially in the pandemic moment of Covid-19 [22]. With that in mind, in this present study, we simulated a situation of droplets and highly contaminated aerosols dispersion in a dental environment. Each of the 12 participants used, in each clinic, a bacterial solution concentration of $1.5 \times 10^8$ CFU of *Lactobacillus casei* Shirota per mL, with a density of $6.75 \times 10^{10}$ CFU. We observed that

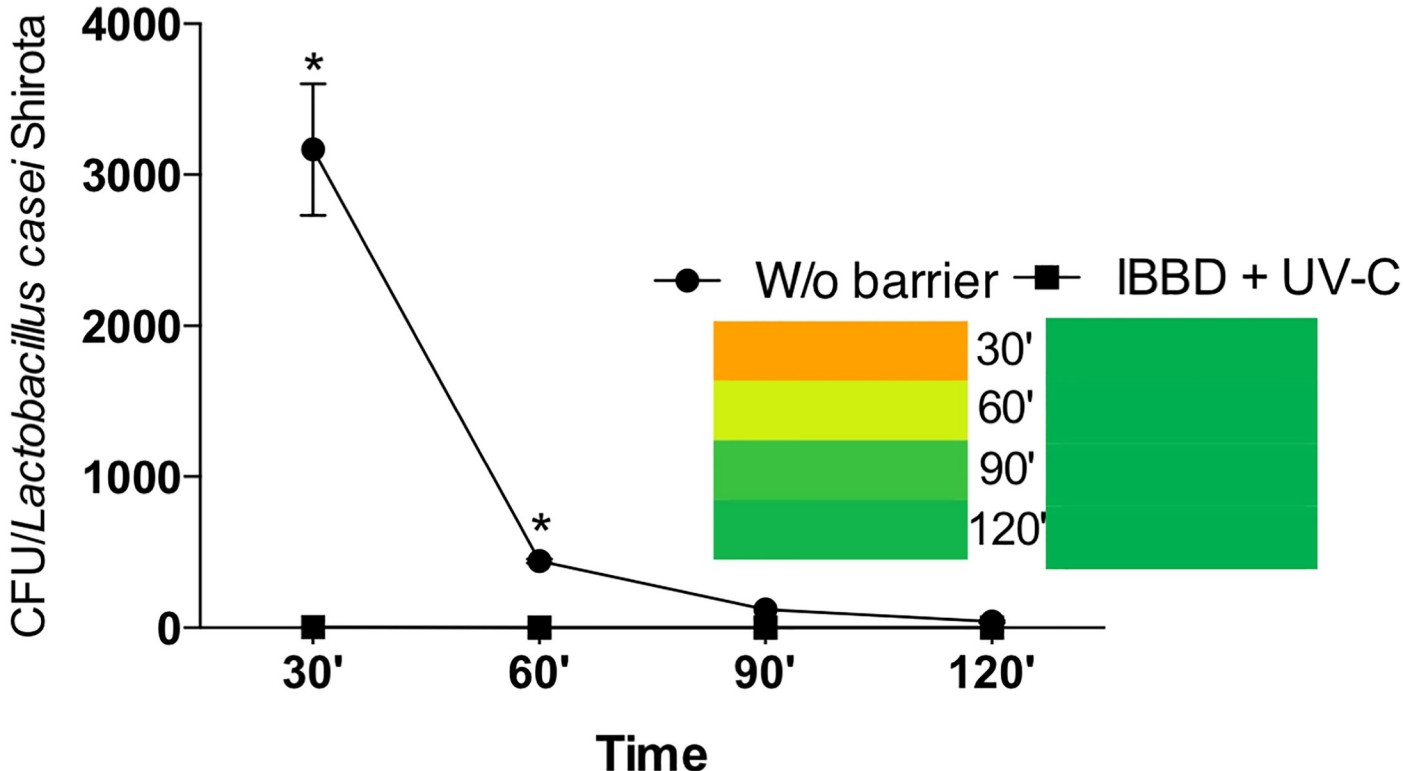

**Fig 3. Sedimentation time in the environments of the positive control group (without barrier) and the IBBD + UV-C group after 30, 60, 90 and 120 minutes of activation of the dental drill containing *L. casei*.** * indicate statistical significance (p < 0.0001). Two-way ANOVA and post hoc analysis using Bonferroni's multiple comparation.

with the use of mechanical barriers (IBBD) or use of physical barriers generated by UV-C, and especially the association of both, they were highly effective in the microbial reduction generated by the dispersion of droplets and aerosols.

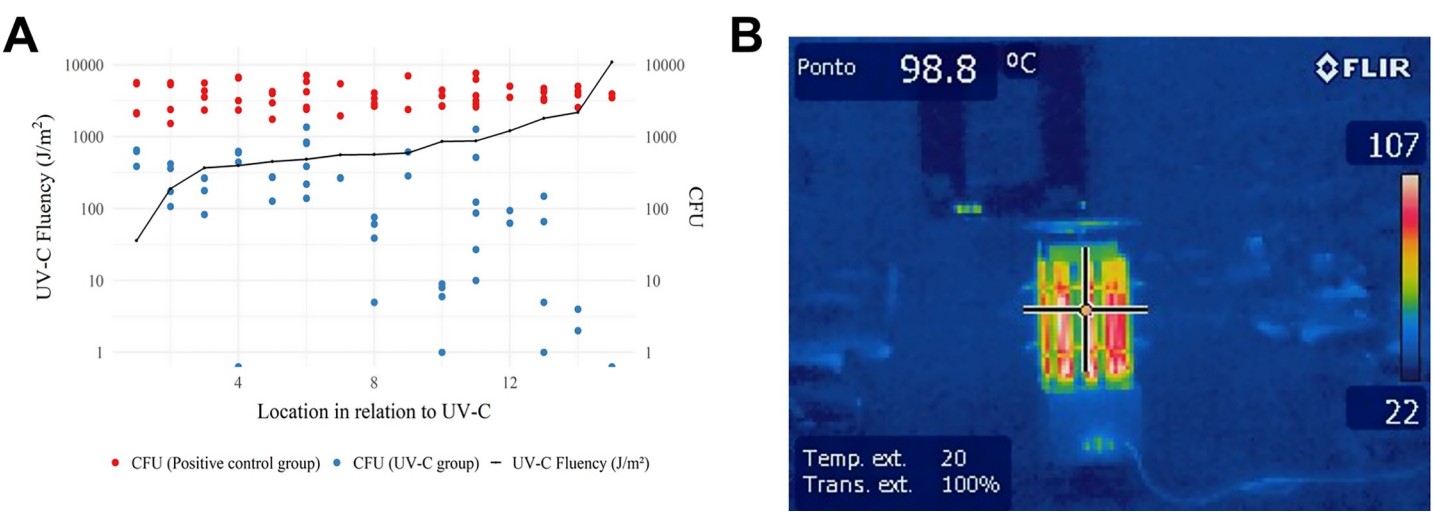

**Fig 4.** Correlation between CFU and UV-C fluency (J/m$^2$) (A) and thermal imaging of UV-C device (B).

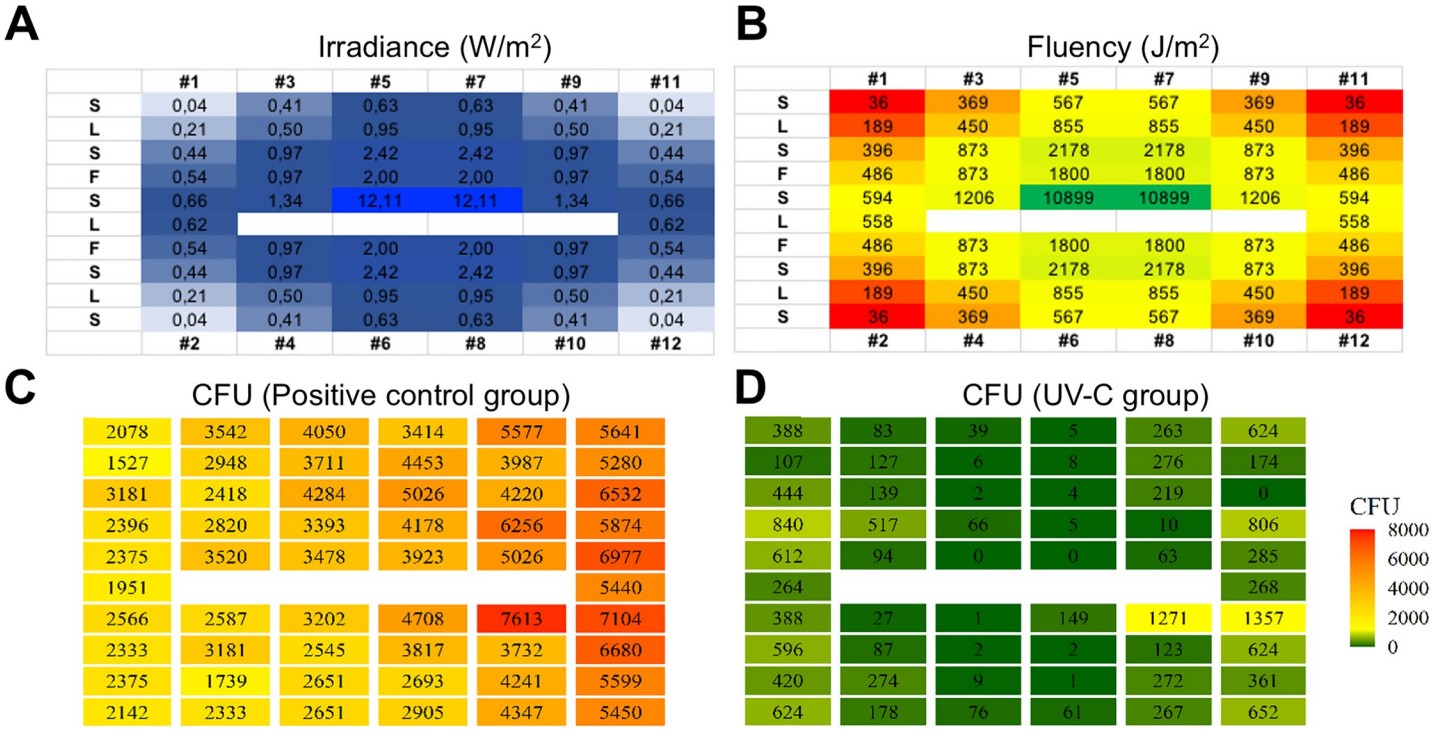

**Fig 5. Data and heat map of irradiance (W/m²) and fluency (J/m²) in the clinic group UV-C.** S = Stand; L = fluorescent lamp fixtures; F = floor; # (number) Position of dental equipment.

In a real situation, PCDP generated using peripheric dental gadgets spread water, saliva and blood that can carry fungi, bacteria and viruses and spread over great distances [5]. Moreover, in a school clinic environment, with several dental equipment being used at the same time, the number of pathogens in the environment multiplies exponentially. Recent studies have related that the larger the enclosed space, the longer the pathogens in the air can settle on the surface [23, 24] hence, traditional disinfection with chemicals is performed, using enzymatic disinfectants for contaminated equipment and surfaces.

Strategies used to reduce the risk of contamination in a health environment have been developed, but at times the evidence has been simulated in a laboratory environment [4, 5, 8, 9, 13, 14]. In the present study we used a real care environment and all variables were considered in the measurements. Among the strategies tested, the Individual Biosafety Barrier in Dentistry (IBBD), which is a metallic device that serves as a support for a disposable plastic barrier that prevents, during patient care, the dispersion of droplets and aerosols in the environment, while allowing good visualization of the operator to perform the procedures. It has already been shown to reduce by up to 95% the dispersion of droplets and dental aerosol [11]. In the present study, with the activation of 12 dental chairs at the same time, only the use of IBBD was able to reduce the CFU count by, on average, 75%.

We also use ultraviolet light technology band C (UV-C), recognized for its ability to kill, or inactivate pathogens [16–20]. Ultraviolet germicidal irradiation (UVGI) is generated by germicidal lamps and can eliminate microorganisms that are in the air or on directly irradiated surfaces. In this study, we used an equipment with 8 mercury vapor lamps of 304μW/cm² of irradiance, without ozone generation. This technology was previously used to control outbreaks of tuberculosis [25], influenza virus H1N1 [26] and recently in the covid-19 pandemic

[27]. In the present study, only UV-C technology reduced environmental contamination by, on average, 93%.

The process of ultraviolet disinfection may involve simple exponential decay, or a more complex function composed of two or more decay, shoulder or delayed response processes and photo-reactivation. The whole process may also be subject to relative moisture effects [17]. Furthermore, the exposure dose itself may be subject to variations of an irregular irradiance field (in the air or on surfaces) and, in the case of air disinfection, there may be irregularities in the airflow caused by the various obstacles within the environment. Each of these components of the disinfection process can be described with a mathematical model, but in the present study, the calculations performed to evaluate irradiance ($W/m^2$) and fluency ($J/m^2$) of UV-C technology were performed in a real clinical setting. The consequent areas of shadow resulting from the positions of dental equipment associated with microbial counts to evaluate PCDP allow a reliable analysis of the capacity of environmental disinfection by UV-C, if parameters such as time, fluency and area are used appropriately.

Associating the two methods of dispersion control of PCDP (IBBD + UV-C), the reduction achieved was, on average, 96%. This result allows us to suggest that both methods, alone or associated, can bring real benefits to health professionals in environments at high risk of cross-contamination, such as dental or hospital environments. Although the studied model used bacterial strain, when they were exposed to 8 95W lamps of 304μW/cm$^2$ of irradiance each, for 15 minutes and with amount of air volume in each clinic of 205.5m$^3$, the elimination of micro-organisms in the form of PCDP was highly effective. The average energy dose for lactobacilli occurs at a relatively low level, with energy doses of 260 and 120 $J/m^2$ [28], and the dose to eliminate the SARS-CoV-2 virus is 16.98 $J/m^2$ (D90) and 33.96 $J/m^2$ (D99) [27].

The UV-C equipment used in the present study generated 12.11 $W/m^2$ and, therefore, it is possible to extrapolate that in addition to the bacteria tested, it would be possible to eliminate viruses such as SARS-CoV-2. For the reduction of 1 log (D90) at 1 meter, it is necessary the time of 1.50 seconds; for the distance of 6 meters (clinic limit) the time required is 50.47 seconds. For the reduction of 2 log (D99), for the same distances, it takes 2.80 seconds and 101 seconds, respectively. It is worth mentioning that, in the studied environment, the extremities of the clinics received the fluency of 36 $J/m^2$ during the 15 minutes of the connected UV-C equipment, which would be, theoretically, sufficient to eliminate the SARS-CoV-2 virus (D99).

Humans produce aerosols continuously through normal breathing [29], however, aerosol production increases during respiratory diseases [30], and individuals infected with SARS-CoV-2 can produce viral aerosols that can remain infectious for long periods of time [1, 7, 8–10, 31].

There are significant gaps in evidence and quality that limit the findings around all aspects of contamination for different procedures. However, in the present study, it can be observed that the association between the methods (IBBD + UV-C) the highest concentration of CFU identified was on the ground, which allows us to emphasize that the health team should not depend only on a single strategy to minimize the risk of contamination. It is necessary to follow all standard precautionary measures, such as the use of Personal Protective Equipment by the team and strictly follow biosafety protocols. Moreover, to reduce the presence of SARS-CoV-2 in the air, room ventilation is strongly recommended, especially in areas where aerosol generation procedures are performed [24, 25].

In addition, it is worth the attention to the high CFU count of the corridor from the droplet dispersion area (clinic without barriers). Measures should be taken to reduce both, the risk of contamination in the contaminated environment, and in environments that give access to critical areas of contamination. Furthermore, we noticed that 30 minutes after the end of the generation of PGDP, high numbers of CFU were found, and from 60 minutes, regardless of the

use of barriers, there is a sharp drop in the number of counts, demonstrating the importance of microorganisms in suspension that can be sources of cross-transmission between individuals. Particles suspended in the air during and after dental care can reach the respiratory tract and connective membranes of dental professionals and patients who will be treated later [32] raising the risk of cross-infection, including the SARS-CoV-2 virus.

Although beyond the scope of this study, future studies should address the effects of moisture and other pathogens, as higher doses of UV-C may be required for the inactivation of other microorganisms. Moreover, because UV irradiation increases mutations, UV irradiation can potentially induce UV-C resistance, as previously reported [33]. However, when the UV dose reaches high enough to kill the entire bacterial population, the emergence of a UV-resistant phenotype can be prevented, indicating that UV-C disinfection is a safe and effective measure to employ in clinical settings without being concerned about the appearance of UV resistance [34]. It is also important to highlight that direct exposure of the skin and eyes to UV-C light can pose a serious health risk, such as corneal irritation and burns [35] and as such, UV-C light should only be used with proper training or where people are not at risk of being exposed. In addition, the lack of a regulatory body that validates UV-C equipment makes evidence-based clinical recommendations and policy decision-making, especially relevant to healthcare, difficult.

Our study has some limitations, including no assessments for potential adverse effects on plastics were conducted. Furthermore, because irradiance and dosages were determined using a single UV-C device, our findings cannot be considered representative of all such devices. Other systems currently available differ in the type and size of UV-C bulbs utilized, the type of reflective surfaces behind bulbs, and methods for monitoring UV-C dosage. We did not evaluate the ability of the UV-C device to reduce bacterial levels on high-touch surfaces or to reduce healthcare-associated infections.

In spite of the existence of several techniques that can reduce the spread of pathogens, the lack of proven effective interventions has allowed the uncontrollable spread of the virus in the human population. Our results show that IBBD and UV-C are powerful tools that can be applied extensively in a wide range of institutions, including hospitals, outpatient clinics and dental offices, to disinfect the potentially contaminated environment, preventing and reducing the transmission of pathogens, including the SARS-CoV-2 virus.

## Supporting information

**S1 Fig. Diagram of the floor plan of the dental clinic with 12 equipment and identification of the sites with the name of the place where the Petri dishes with MRS agar medium were positioned in the tests.** On the surface (Stand), green circles; On the fluorescent lamp fixture (LF), orange circles; On the floor (Floor), blue circles; Position of operators for activation of dental drill (aerosol activation), red squares.
(TIF)

**S2 Fig. Twelve volunteers were positioned in the same dental chair position, on the right side of work, in the 4 different clinics and all activated dental drill at the same time.** After activation, the Petri dishes were opened and remained open for 15 minutes in the pre established position. Clinic Positive Control Group (no barriers) (A); Clinic Individual Biosafety Barrier in Dentistry (IBBD) Group (B); Clinic UV-C Group (C) and; Clinic IBBD + UV-C Group (D).
(TIF)

**S3 Fig. Individual Biosafety Barrier in Dentistry (IBBD).** Protection barrier against droplets and aerosol is made using a metal support, with a 30 cm ring and covered by a disposable 30

microns thickness PVC film measuring approximately 1.5 x 1.5 m (A); Ultraviolet-C Device Mobile Disinfection Unit (UMDUV 2.0). The equipment is composed of 8 UVGI lamps of 95W of low-pressure mercury with $304\mu W/cm^2$ irradiance, without ozone generation (B). (TIF)

**S4 Fig.** Mass (in grams) of the bacterial solution after activation of dental drill refrigeration (A) in the packages and dispersed in the environment (B) (p >0.05). (TIF)

**S5 Fig.** Scheme 3 D with Heat map of the results obtained from CFU counts in clinics: positive control (A); IBBD; UV-C; and IBBD + UV-C. The mean difference between the positive control group and the IBBD group was, on average, 75% (B). When using UV-C device, average CFU counts had, on average, a 93% (C) and 96% reduction in IBBD+UV-C (D). (TIF)

**S1 Table.** (DOCX)

## Acknowledgments

The authors acknowledge the students Ana Paula Machado Stefanini, Fabiano Augusto dos Santos Janisch, Isa Furlan, Isabela França Moreno, João Pedro Andrade Silva, João Pedro Grandini Zeferino, Laira Lourenço Chegure, Letícia Carvalho Dezolt, Nathália Ribeiro Brochado de Almeida and Professor Aguinaldo Silva Garcez for their participation in the microbiological tests. The authors also acknowledge the technical assistance of Thiago Almeida and Gilca Saba in the laboratory of microbiology and the engineer Dayane Pereira Lima Santos for helping with the elaboration of the 3D figures at Faculdade São Leopoldo Mandic. The authors should also acknowledge Dr. Elizabeth Menzl for promptly volunteering to review this manuscript regarding its English language content and Dr. Rafael Bovi Ambrosano for helping with the statistical analysis of the data.

## Author Contributions

**Conceptualization:** Victor Angelo Martins Montalli, Marcelo Henrique Napimoga.

**Data curation:** Victor Angelo Martins Montalli, Milenna de Figueiredo Torres, Oscar de Figueiredo Torres Junior, Dienne Hellen Moutinho De Vilhena.

**Formal analysis:** Victor Angelo Martins Montalli, Marcelo Henrique Napimoga.

**Investigation:** Victor Angelo Martins Montalli, Patrícia Rejane de Freitas, Milenna de Figueiredo Torres, Dienne Hellen Moutinho De Vilhena.

**Methodology:** Victor Angelo Martins Montalli, Patrícia Rejane de Freitas, Milenna de Figueiredo Torres, Oscar de Figueiredo Torres Junior, Dienne Hellen Moutinho De Vilhena, Marcelo Henrique Napimoga.

**Project administration:** Victor Angelo Martins Montalli, Marcelo Henrique Napimoga.

**Resources:** José Luiz Cintra Junqueira.

**Supervision:** Victor Angelo Martins Montalli.

**Validation:** Victor Angelo Martins Montalli, Oscar de Figueiredo Torres Junior, Dienne Hellen Moutinho De Vilhena, José Luiz Cintra Junqueira.

**Writing – original draft:** Victor Angelo Martins Montalli, Marcelo Henrique Napimoga.

**Writing – review & editing:** Victor Angelo Martins Montalli, José Luiz Cintra Junqueira, Marcelo Henrique Napimoga.

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
