## [Decision Letter · Decision Letter 0]

10 Jun 2021

PONE-D-21-11593

Biosafety devices to control the spread of potentially contaminated dispersion particles. New associated strategies for health environments.

PLOS ONE

Dear Dr. Montalli,

Thank you for submitting your manuscript to PLOS ONE. After careful consideration, we feel that it has merit but does not fully meet PLOS ONE’s publication criteria as it currently stands. Therefore, we invite you to submit a revised version of the manuscript that addresses the points raised during the review process.

We look forward to receiving your revised manuscript.

Kind regards,

Amitava Mukherjee, ME, Ph.D.

Academic Editor

PLOS ONE

Journal Requirements:

"Conselho Nacional de Desenvolvimento Científico e Tecnológico (CNPq)

(Research Productivity Fellowships was awarded to MHN), Coordenação de

Aperfeiçoamento de Pessoal de Nível Superior, CAPES (#001)"

We note that one or more of the authors are employed by a commercial company: UVCtec Company.

3.1. Please provide an amended Funding Statement declaring this commercial affiliation, as well as a statement regarding the Role of Funders in your study. If the funding organization did not play a role in the study design, data collection and analysis, decision to publish, or preparation of the manuscript and only provided financial support in the form of authors' salaries and/or research materials, please review your statements relating to the author contributions, and ensure you have specifically and accurately indicated the role(s) that these authors had in your study. You can update author roles in the Author Contributions section of the online submission form.

3.2. Please also provide an updated Competing Interests Statement declaring this commercial affiliation along with any other relevant declarations relating to employment, consultancy, patents, products in development, or marketed products, etc.  

4. Please include your tables as part of your main manuscript and remove the individual files. Please note that supplementary tables should remain uploaded as separate "supporting information" files.

Reviewers' comments:

Reviewer's Responses to Questions

**Comments to the Author**

1. Is the manuscript technically sound, and do the data support the conclusions?

Reviewer #1: Yes

Reviewer #2: Partly

2. Has the statistical analysis been performed appropriately and rigorously? 

Reviewer #1: Yes

Reviewer #2: Yes

3. Have the authors made all data underlying the findings in their manuscript fully available?

Reviewer #1: Yes

Reviewer #2: Yes

4. Is the manuscript presented in an intelligible fashion and written in standard English?

Reviewer #1: No

Reviewer #2: No

5. Review Comments to the Author

Reviewer #1: The study has been rigously performed and reported. The paper needs to be proof read. Grammatical errors have been noticed, 98-102; 279-81. Statistical equations could be elaborated. Limitations of the study need to be addressed.

Reviewer #2: It is an important concern about the aerosol spread in dental therapy and operation especially in the background of the COVID-19 pandemic. I commend the authors’ effort on experimental investigation on airborne bioaerosol control during a simulation drill operation. But there are still important issues which I strongly recommend the authors think about and make improvement.

Major issue 1: Organization of the paper

Comment 1. The organization and the citation of the figures in this manuscript is really confusing to the readers. Just for two examples:

a) The citation of Figure 1 in Line 146 and Line 199 indicate the figure should be the illustration of the test environment and sampling location. But the Figure 1 is actually the CFU measurement results.

b) Both Figure 4 and Figure 4(supplementary) cannot support the citation in Line 295 to Line 299 of the manuscript.

Comment 2. Some experiments mentioned in material and methods. But the corresponding results and discussion are missing. For an example, I did not find the results and discussion about PCDP suspension in the dental clinical room and presence measurement outside the clinical room.

Major issue 2: Validation of the experiment. And Connection between the object and supporting data

Comment 3. The topic of this manuscript is about the control the spread of bioaerosols in health environment. But the experiments were just actually designed to validate the bioaerosol concentration control. The emission sources were placed on all available dental chairs rather than part of the chairs or maybe just one chair. The results of the measurement outside the clinic room are missing. It is impossible for readers to evaluate the control effect of bio-particles spread of the measures mentioned in the paper. On the other hand, the authors should also make introductions on if such strategies or measures could be applied in other health care environment as specified in the tittle.

Comment 4. In the negative control group (Line 279 to 284), the petri dishes indicated wide contamination. The largest observation is 11.3 (6.1) for IBBD group. It is not normal according to my previous research experience and indicate the dishes are not well protected or the preparation did not fulfil the requirements of aseptic operation.

Major issue 3: Limitation of the biosafety risk of UV technology

Comment 5. UV technology is one of most traditional, easily available and widely used disinfection technology. But there are also concerns about the biosafety risk especially for short term radiation, such as 15min in this manuscript. Major concerns include:

a) Variation might happen to the pathogens after UV radiation.

b) The biggest concern is about the pathogen variation on the resistance for antibiotic.

According to a 12 months airborne bacterial monitoring in a hospital in 2004 in Guangxi, China (Yanhua Chen, Hui Li, Yiping Lu and Lingsha Huang, Distribution of bacteriological spectrum in the air of hospital and analysis of their drug resistance, Chinese Journal of Antibiotics, 2006, 31(008):505-506 (in Chinese)):

a) 450 bacterium strains were collected during 833 samplings in 90 locations

b) 72.0% show resistance to penicillin

c) 76.9% show resistance to ampicillin

d) Above 40% show resistance to new antibiotics such as cephalosporins and quinolones

e) The drug-resistance rate in the hospital is much higher than the other monitoring results in communities, which is believed highly relatedly with the wide use the UV disinfection instruments in hospitals in underdeveloped area

Another research published in 2010 (Yan Wang, Yao Zhou and Yinghua Zhang et al., Impact of Ultraviolet Irradiation on Bacterial Drug Resistance, Chinese Journal of Nosocomiology, 2010, 020(021):3355-3356,3363. (in Chinese)) indicate obvious change drug resistance of Serratia marcescens on amoxicillin, aztreonam and cefuroxime after short term UV irradiation and generation multiplication.

I would recommend that the authors make necessary analysis about the minimum dose of the UV radiation, and consider the possibility of extending the radiation time. Or list the related risk and concerns.

Major issue 4: English expression of the paper

Comment 6. I am not a native English speaker. But there are obvious errors in some sentences and expressions such as “The radiometer was operated with specified precision at room temperature of 22°C to 53% relative humidity” in Line 215 to 216. These would cause confusion and misunderstanding of readers. So, I recommend the authors re-check the expression and fix all errors.

6. PLOS authors have the option to publish the peer review history of their article (what does this mean?). If published, this will include your full peer review and any attached files.

Reviewer #1: **Yes: **Bhanu Lakhani

Reviewer #2: No

---

## [Author Response · Author response to Decision Letter 0]

20 Jun 2021

June 20th 2021.

Dr. Joerg Heber

Editor-in-Chief, Plos One

 Please find enclosed a copy of our reviewed manuscript PONE-D-21-11593, "Biosafety devices to control the spread of potentially contaminated dispersion particles. New associated strategies for health environments". We would like to thank you and the reviewers for all constructive comments and we are certain that they were key to improve the manuscript. We accepted most of the suggestions and added information to the text to clarify the points raised by the referees. All changes performed in the manuscript were highlighted in red. Our responses to the individual comments are delineated below. We hope that these corrections will satisfy the reviewer’s comments, and look forward to hearing from you.

Sincerely yours,

Journal Requirements:

A: Thank you for your suggestion. We have included additional information in the manuscript in order to strengthen the transmission. Please see new submitted version of the manuscript. 

"Conselho Nacional de Desenvolvimento Científico e Tecnológico (CNPq)

(Research Productivity Fellowships was awarded to MHN), Coordenação de

Aperfeiçoamento de Pessoal de Nível Superior, CAPES (#001)"

A: Thank you for your careful reading. We have removed the funding from the acknowledgment section and updated the Funding statement. We believe it is important to highlight the unspecific funding because the funding described (CAPES #001) refers to the www.periodicos.capes.gov.br which is the Brazilian national electronic library consortium for science and technology, providing full access to the content of more than 10,000 journals. Since only universities which possess master and PhD programs have access of this consortium, we provide this recognition. On the other hand, the CNPq is an individual fellowship granted to Dr. Marcelo Henrique Napimoga. Thus, none of the funding named here was used to support the current experiments.

We note that one or more of the authors are employed by a commercial company: UVCtec Company.

3.1. Please provide an amended Funding Statement declaring this commercial affiliation, as well as a statement regarding the Role of Funders in your study. If the funding organization did not play a role in the study design, data collection and analysis, decision to publish, or preparation of the manuscript and only provided financial support in the form of authors' salaries and/or research materials, please review your statements relating to the author contributions, and ensure you have specifically and accurately indicated the role(s) that these authors had in your study. You can update author roles in the Author Contributions section of the online submission form.

A: Thank you for your careful reading. Mr. Oscar F. Torres-Junior who has affiliation with UVCtec company helped with the calculation of UV-C radiation in the studied environment as already described in the first submission. We have updated the Funding statement to describe that UVCtec company is a startup and the equipment UMDUV 2.0 is a Minimum Viable Product (MVP) prototype. Mr. Oscar Torres-Junior has support in the form of salary. None of the other authors has received any funding or has any commercial interest with UVCTec. 

3.2. Please also provide an updated Competing Interests Statement declaring this commercial affiliation along with any other relevant declarations relating to employment, consultancy, patents, products in development, or marketed products, etc. 

A: The conflict of interest already provide the information requested. We have included that Mr. Oscar Torres-Junior has support in the form of salary from UVCTec, although this does not alter our adherence to PLOS ONE policies on sharing data and materials.

4. Please include your tables as part of your main manuscript and remove the individual files. Please note that supplementary tables should remain uploaded as separate "supporting information" files.

A: Thank you for your careful reading. We have provided the corrections. 

 

Reviewers' comments:

Reviewer #1: The study has been rigously performed and reported. The paper needs to be proof read. Grammatical errors have been noticed, 98-102; 279-81. Statistical equations could be elaborated. Limitations of the study need to be addressed.

A: Thank you for your careful reading. Below the new information included in the revised manuscript. Please see new submitted version of the manuscript.

98-102:

“Among microorganisms that are potentially contagious to health professionals operating near the face and oral cavity, especially when PCDP is generated [10], are hepatitis B virus, HIV (human immunodeficiency virus).as well as SARS-CoV-2 (Severe Acute Respiratory Syndrome coronavirus 2). The latter can remain infectious in aerosols for long periods, even when water evaporates, and particles that settle on surfaces can remain infectious for up to 72 h [8, 9].”

279-81:

“From the Petri dishes arranged in dental clinics before the beginning of the tests, the negative control group had low CFU counts. The group without barriers had mean (standard deviation) of 1.3 (1.0) CFU; the IBBD group 11.3 (6.1); the UV-C group 1.0 (0.8) and in the IBBD + UV-C group 2.3 (2.1), with no statistically significant difference (p>0.05) between the dental clinics used for the experiments.”

Limitations of the study was addressed in the discussion:

“Our study has some limitations, including no assessments for potential adverse effects on plastics were conducted. Furthermore, because irradiance and dosages were determined using a single UV-C device, our findings cannot be considered representative of all such devices. Other systems currently available differ in the type and size of UV-C bulbs utilized, the type of reflective surfaces behind bulbs, and methods for monitoring UV-C dosage. We did not evaluate the ability of the UV-C device to reduce bacterial levels on high-touch surfaces or to reduce healthcare-associated infections.”

Reviewer #2: It is an important concern about the aerosol spread in dental therapy and operation especially in the background of the COVID-19 pandemic. I commend the authors’ effort on experimental investigation on airborne bioaerosol control during a simulation drill operation. But there are still important issues which I strongly recommend the authors think about and make improvement.

Major issue 1: Organization of the paper

Comment 1. The organization and the citation of the figures in this manuscript is really confusing to the readers. Just for two examples:

a) The citation of Figure 1 in Line 146 and Line 199 indicate the figure should be the illustration of the test environment and sampling location. But the Figure 1 is actually the CFU measurement results.

b) Both Figure 4 and Figure 4(supplementary) cannot support the citation in Line 295 to Line 299 of the manuscript.

A: Thank you for your careful reading. We have included additional information in the citation of the figures. Please see new submitted version of the manuscript.

Comment 2. Some experiments mentioned in material and methods. But the corresponding results and discussion are missing. For an example, I did not find the results and discussion about PCDP suspension in the dental clinical room and presence measurement outside the clinical room.

A: The data regarding this issue was provided in the first submission as described below:

Results section

To evaluate the distance that PCDP can reach from the generating source, new Petri dishes were positioned in the corridor that connect the clinics to the end of the group test without barriers. As shown in Fig 2E, it was observed the growth of 903 (28) CFU in the plates positioned 1m from the input (yellow box), 3044 (64) CFU at 5m distance (dark orange box) and 1966 (42) CFU 10 m away from the generating source (light orange box). Schematic 3D data are presented in S5 Fig.

Discussion section

In addition, it is worth the attention to the high CFU count of the corridor from the droplet dispersion area (clinic without barriers). Measures should be taken to reduce both, the risk of contamination in the contaminated environment, and in environments that give access to critical areas of contamination. Furthermore, we noticed that 30 minutes after the end of the generation of PGDP, high numbers of CFU were found, and from 60 minutes, regardless of the use of barriers, there is a sharp drop in the number of counts, demonstrating the importance of microorganisms in suspension that can be sources of cross-transmission between individuals. Particles suspended in the air during and after dental care can reach the respiratory tract and connective membranes of dental professionals and patients who will be treated later [32] raising the risk of cross-infection, including the SARS-CoV-2 virus.

Major issue 2: Validation of the experiment. And Connection between the object and supporting data

A: The current issue raised by the referee is not a question which we can properly answer. As demonstrated, all experiments are well designed and described, as well as the statistical analysis. 

Comment 3. The topic of this manuscript is about the control the spread of bioaerosols in health environment. But the experiments were just actually designed to validate the bioaerosol concentration control. The emission sources were placed on all available dental chairs rather than part of the chairs or maybe just one chair. The results of the measurement outside the clinic room are missing. It is impossible for readers to evaluate the control effect of bio-particles spread of the measures mentioned in the paper. On the other hand, the authors should also make introductions on if such strategies or measures could be applied in other health care environment as specified in the tittle.

A: Thank you for your careful reading. As described below the results of outside clinic is already provided in the results section as well as in the discussion. In the current manuscript, we have designed an experimental design to spread a high density of contaminated dispersion particles in a university dental environment which has 12 dental chairs. This environmental condition with a high concentration of microorganism dispersion allowed us to test whether the barriers used for the control (individual biosafety barrier in dentistry) and UV-C light would be able to prevent dispersion. 

Since we tested the dispersion of microorganisms using 12 dental chairs at the same time, and Petri dishes were placed in various places within the same room, totaling 56 dishes in each room, including the light fixtures on the room ceiling, we can see the potential for dispersion generated in our proposed methodology, as well as the ability to avoid such dispersion using the proposed barriers. Thus, given such a great challenge induced by this methodology, we believe it is possible to extrapolate that the measures tested here can be extrapolated to other health environments.

We have included an additional paragraph in the introduction to expand the overview of the possible use of the strategies in other health environments.

“Ultraviolet disinfection technology can be used to supplement manual cleaning, and recently it has become an acceptable method of no-touch disinfection within healthcare facilities and is currently routinely used in disinfecting the hospital environment with HAI reduction having been demonstrated in previous studies [15-20].”

Comment 4. In the negative control group (Line 279 to 284), the petri dishes indicated wide contamination. The largest observation is 11.3 (6.1) for IBBD group. It is not normal according to my previous research experience and indicate the dishes are not well protected or the preparation did not fulfil the requirements of aseptic operation.

A: Thank you for your careful reading. The idea of this information is to demonstrate that prior to the beginning of the experiment there was no significant contamination of Lactobacillus casei in the test environment. However, to avoid misunderstanding we renamed this test by removing the negative control word.

Major issue 3: Limitation of the biosafety risk of UV technology

Comment 5. UV technology is one of most traditional, easily available and widely used disinfection technology. But there are also concerns about the biosafety risk especially for short term radiation, such as 15min in this manuscript. Major concerns include:

a) Variation might happen to the pathogens after UV radiation.

b) The biggest concern is about the pathogen variation on the resistance for antibiotic.

According to a 12 months airborne bacterial monitoring in a hospital in 2004 in Guangxi, China (Yanhua Chen, Hui Li, Yiping Lu and Lingsha Huang, Distribution of bacteriological spectrum in the air of hospital and analysis of their drug resistance, Chinese Journal of Antibiotics, 2006, 31(008):505-506 (in Chinese)):

a) 450 bacterium strains were collected during 833 samplings in 90 locations

b) 72.0% show resistance to penicillin

c) 76.9% show resistance to ampicillin

d) Above 40% show resistance to new antibiotics such as cephalosporins and quinolones

e) The drug-resistance rate in the hospital is much higher than the other monitoring results in communities, which is believed highly relatedly with the wide use the UV disinfection instruments in hospitals in underdeveloped area

Another research published in 2010 (Yan Wang, Yao Zhou and Yinghua Zhang et al., Impact of Ultraviolet Irradiation on Bacterial Drug Resistance, Chinese Journal of Nosocomiology, 2010, 020(021):3355-3356,3363. (in Chinese)) indicate obvious change drug resistance of Serratia marcescens on amoxicillin, aztreonam and cefuroxime after short term UV irradiation and generation multiplication.

I would recommend that the authors make necessary analysis about the minimum dose of the UV radiation, and consider the possibility of extending the radiation time. Or list the related risk and concerns.

A: Thank you for your careful reading. We have included additional information about the possible UV-C use and UV resistance microorganisms.

“Moreover, because UV irradiation increases mutations, UV irradiation can potentially induce UV-C resistance, as previously reported [33]. However, when the UV dose reaches high enough to kill the entire bacterial population, the emergence of a UV-resistant phenotype can be prevented, indicating that UV-C disinfection is a safe and effective measure to employ in clinical settings without being concerned about the appearance of UV resistance [34].”

33. Shoults DC, Ashbolt NJ. Decreased efficacy of UV inactivation of Staphylococcus aureus after multiple exposure and growth cycles. Int J Hyg Environ Health. 2019 Jan;222(1):111-116. doi: 10.1016/j.ijheh.2018.08.007.

34. Choi H, Chatterjee P, Hwang M, Stock EM, Lukey JS, Zeber JE, et al. Can multidrug-resistant organisms become resistant to ultraviolet (UV) light following serial exposures? Characterization of post-UV genomic changes using whole-genome sequencing. Infect Control Hosp Epidemiol. 2021 Mar 22:1-7. doi: 10.1017/ice.2021.51. 

Major issue 4: English expression of the paper

Comment 6. I am not a native English speaker. But there are obvious errors in some sentences and expressions such as “The radiometer was operated with specified precision at room temperature of 22°C to 53% relative humidity” in Line 215 to 216. These would cause confusion and misunderstanding of readers. So, I recommend the authors re-check the expression and fix all errors.

A: Thank you for your careful reading. Below the new information included in the revised manuscript. Please see new submitted version of the manuscript.

215 to 216:

“The radiometer was operated according to its specification at room temperature of 22°C at 53% of air relative humidity.”

---

## [Decision Letter · Decision Letter 1]

19 Jul 2021

Biosafety devices to control the spread of potentially contaminated dispersion particles. New associated strategies for health environments.

PONE-D-21-11593R1

Dear Dr. Montalli,

We’re pleased to inform you that your manuscript has been judged scientifically suitable for publication and will be formally accepted for publication once it meets all outstanding technical requirements.

Kind regards,

Amitava Mukherjee, ME, Ph.D.

Academic Editor

PLOS ONE

Additional Editor Comments (optional):

Reviewers' comments:

Reviewer's Responses to Questions

**Comments to the Author**

1. If the authors have adequately addressed your comments raised in a previous round of review and you feel that this manuscript is now acceptable for publication, you may indicate that here to bypass the “Comments to the Author” section, enter your conflict of interest statement in the “Confidential to Editor” section, and submit your "Accept" recommendation.

Reviewer #2: All comments have been addressed

2. Is the manuscript technically sound, and do the data support the conclusions?

Reviewer #2: Yes

3. Has the statistical analysis been performed appropriately and rigorously? 

Reviewer #2: Yes

4. Have the authors made all data underlying the findings in their manuscript fully available?

Reviewer #2: Yes

5. Is the manuscript presented in an intelligible fashion and written in standard English?

Reviewer #2: Yes

6. Review Comments to the Author

Reviewer #2: (No Response)

7. PLOS authors have the option to publish the peer review history of their article (what does this mean?). If published, this will include your full peer review and any attached files.

Reviewer #2: No

---

## [Editor Report · Acceptance letter]

18 Aug 2021

PONE-D-21-11593R1 

Biosafety devices to control the spread of potentially contaminated dispersion particles. New associated strategies for health environments. 

Dear Dr. Montalli:

I'm pleased to inform you that your manuscript has been deemed suitable for publication in PLOS ONE. Congratulations! Your manuscript is now with our production department. 

Kind regards, 

on behalf of

Professor Dr. Amitava Mukherjee 

Academic Editor

PLOS ONE